# NRAS is unique among RAS proteins in requiring ICMT for trafficking to the plasma membrane

Ian M Ahearn[1,2,3] , Helen R Court[2] , Farid Siddiqui[4] , Daniel Abankwa[4,5] , Mark R Philips[2]

**Isoprenylcysteine carboxyl methyltransferase (ICMT) is the third of three enzymes that sequentially modify the C-terminus of CaaX proteins, including RAS. Although all four RAS proteins are substrates for ICMT, each traffics to membranes differently by virtue of their hypervariable regions that are differentially palmitoylated. We found that among RAS proteins, NRAS was unique in requiring ICMT for delivery to the PM, a consequence of having only a single palmitoylation site as its secondary affinity module. Although not absolutely required for palmitoylation, acylation was diminished in the absence of ICMT. Photoactivation and FRAP of GFP-NRAS revealed increase flux at the Golgi, independent of palmitoylation, in the absence of ICMT. Association of NRAS with the prenyl-protein chaperone PDE6δ also required ICMT and promoted anterograde trafficking from the Golgi. We conclude that carboxyl methylation of NRAS is required for efficient palmitoylation, PDE6δ binding, and homeostatic flux through the Golgi, processes that direct delivery to the plasma membrane.**

## Introduction

The regulation and function of RAS proteins are coordinated at the cytosolic face of cellular membranes. As such, the biochemical mechanisms that support RAS membrane association regulate its biological functions (Willumsen et al, 1984; Gutierrez et al, 1989; Hancock et al, 1989). To achieve membrane affinity, all RAS proteins undergo sequential post-translational modifications of their C-terminal CaaX motifs that include farnesylation, endoproteolysis, and carboxyl methylation (Wright & Philips, 2006). Farnesylation is the covalent conjugation of a 15-carbon polyisoprenoid lipid to the CaaX cysteine via a thioether bond, a reaction catalyzed by farnesyltransferase (FTase) (Schafer et al, 1990). Farnesylation is irreversible and alone imparts weak affinity for membranes, particularly the cytosolic face of the ER (Choy et al, 1999). At the ER, farnesylated RAS undergoes endoproteolytic

cleavage of the -aaX tripeptide, which is catalyzed by the RAS converting enzyme (RCE1) (Boyartchuk et al, 1997; Schmidt et al, 1998; Freije et al, 1999; Otto et al, 1999). The last step of CaaX processing is the methylesterification of the terminal $\alpha$-carboxylate of the prenylcysteine by the ER-restricted enzyme isoprenylcysteine carboxyl methyltransferase (ICMT) (Gutierrez et al, 1989; Dai et al, 1998). Unlike farnesylation and proteolysis, carboxyl methylation of RAS is reversible (Chelsky et al, 1985) and, therefore, its contribution to membrane binding is dynamic.

In the absence of secondary membrane-targeting signals, CaaX-modified RAS is directed to the ER but is unable to reach the plasma membrane (PM) (Choy et al, 1999; Apolloni et al, 2000; Tsai et al, 2015). The second signals present in all RAS isoforms that support PM trafficking are encoded by distinct C-terminal amino acid sequences immediately upstream of the CaaX sequence. These sequences combined with the CaaX motif are designated the hypervariable regions (HVRs). KRAS4B harbors a polybasic amino acid motif upstream of the prenylcysteine which allows for an electrostatic interaction with negatively charged phospholipids of the inner leaflet of the PM (Hancock et al, 1990). In the case of HRAS, NRAS, and KRAS4A, the second signals for PM trafficking are generated by palmitoylation of either 1 (NRAS, KRAS4A) or 2 (HRAS) cysteines in the HVR (Hancock et al, 1989). In addition to palmitoylation, KRAS4A has a polybasic motif weaker than that of KRAS4B but capable of functioning in conjunction with the palmitate modification (Tsai et al, 2015). Thus, RAS isoforms undergo both common and unique post-translational modifications that result in distinct subcellular trafficking and compartmentalization. These isoform-specific mechanisms of RAS trafficking have the potential to diversify the biologic function of different RAS isoforms (Amendola et al, 2019).

Although farnesylation alone is insufficient to direct RAS to the PM, it is required for all subsequent modifications and to cooperate effectively with intrinsic electrostatic affinities. Therefore, RAS proteins that cannot be farnesylated remain soluble. Although membrane association of RAS does not absolutely require RCE1 and ICMT, both enzymes increase membrane affinity and loss of either RCE1 or ICMT results in partial mislocalization of a subset of

[1]The Ronald O Perelman Department of Dermatology, New York University Grossman School of Medicine, New York, NY, USA  [2]The Perlmutter Cancer Center, New York University Langone Medical Center, New York, NY, USA  [3]Veterans Affairs New York Harbor Healthcare System, Manhattan Campus, New York, NY, USA  [4]Turku Bioscience Centre, University of Turku and Åbo Akademi University, Turku, Finland  [5]Cancer Cell Biology and Drug Discovery Group, Department of Life Sciences and Medicine, University of Luxembourg, Esch-sur-Alzette, Luxembourg

Correspondence: ian.ahearn@nyulangone.org

farnesylated small GTPases, including RAS (Bergo et al, 2000, 2002; Michaelson et al, 2005). Whereas knockout of CaaX processing enzymes is embryonic lethal, conditional loss of *RCE1* and *ICMT* have been studied in genetically engineered mouse models of *RAS*-driven neoplasia (Wahlstrom et al, 2007, 2008; Court et al, 2013). In these models, whereas loss of *ICMT* was able to ameliorate *KRAS*-driven myeloproliferative disease (Wahlstrom et al, 2008), loss of *ICMT* accelerated disease progression of *KRAS*-driven pancreatic neoplasia (Court et al, 2013). These discrepant results illustrate that the role of ICMT in neoplasia is context dependent and likely explained by the fact that ICMT methylesterifies hundreds of CaaX protein substrates. Indeed, the paradoxical effect of ICMT deficiency in a *KRAS*-driven model of pancreatic cancer was attributed to a requirement for ICMT for RAB-mediated NOTCH1 trafficking (Hanlon et al, 2010; Court et al, 2017).

To date, studies designed to validate ICMT inhibition as a strategy to block oncogenic RAS have focused on KRAS (Wahlstrom et al, 2008; Court et al, 2013; Lau et al, 2017). Interestingly, a CRISPR-based genomic screen to identify novel targets in myeloid leukemia identified ICMT in cell lines where *NRAS* mutations were present (Wang et al, 2017), highlighting the possibility that ICMT may impact the signaling capacity of RAS isoforms differently.

NRAS is the RAS isoform most commonly mutated in cutaneous melanoma and many hematologic cancers (Prior et al, 2012). For NRAS, the second signal that is required for transit to the PM consists of a single cysteine (cys181) that undergoes *S*-palmitoylation. Palmitoylation is the covalent attachment of a 16-carbon saturated palmitoyl lipid to the sulfhydryl group of a cysteine and is catalyzed by acyl transferases resident on the Golgi (Resh, 1999, 2016). Palmitoylation increases the affinity of NRAS at the Golgi and thereby promotes sorting to vesicles that travel to the PM (Shahinian & Silvius, 1995; Rocks et al, 2010). Like methylation, palmitoylation is reversible (Magee et al, 1987) and, for NRAS, depalmitoylation is enzymatically catalyzed by serine hydrolases, including ABHD17A (Martin et al, 2011; Lin & Conibear, 2015). At present, a leading conceptual model posits that post-prenylation subcellular compartmentalization of NRAS is established predominantly through the restriction of the palmitoylation machinery at the Golgi and this, in turn, promotes vesicular trafficking to the PM (Rocks et al, 2005). Efficient depalmitoylation at the PM and elsewhere results in a rapid, fluid phase transport of depalmitoylated NRAS back to the Golgi where another acylation cycle ensues (Goodwin et al, 2005). Interestingly, although palmitoylation is required only for PM binding and not for association with endomembranes, inhibition of palmitoylation was found to abrogate the transforming potential of oncogenic NRAS in mouse models of leukemia (Cuiffo & Ren, 2010).

In addition to CaaX processing and palmitoylation, the regulation of NRAS subcellular compartmentalization is also regulated by interactions with prenyl-binding chaperones, including PDE6$\delta$ and VPS35 (Nancy et al, 2002; Chandra et al, 2011; Zhou et al, 2016). Binding to PDE6$\delta$ occurs preferentially with unpalmitoylated NRAS (Chandra et al, 2011) whereas binding to KRAS4B, which is not palmitoylated, is favored by carboxyl methylation (Dharmaiah et al, 2016). HRAS does not bind PDE6$\delta$ (Chandra et al, 2011; Tsai et al, 2015). Here, we show that NRAS is unique among RAS proteins in

requiring carboxyl methylation for delivery to the PM, which is a consequence of diminished dwell time of unmethylated NRAS on Golgi membranes.

# Results

## ICMT is required for NRAS expression on the PM

To determine the role of ICMT in RAS localization in human tumor cells we targeted the *ICMT* locus in SKMEL28 melanoma cells using lentiviral delivery of CRISPR/Cas9 and either a control single guide RNA (sgRNA) targeting Tomato fluorescent protein (sg[Tom]) or a sgRNA targeting exon 1 of *ICMT* (sg[ICMT]). After selection, we confirmed the loss of ICMT protein expression by immunoblot of whole-cell lysates (Fig 1A). Functional loss of ICMT was confirmed by measuring the enzymatic activity in membrane fractions derived from these cells (Fig 1B) as well as in intact cells by metabolically labelling with [$^3$H]-ʟ-methyl methionine and measuring incorporation of alkaline-labile $^3$H into GFP-NRAS (Fig 1C) (Choy & Philips, 2000). The prenylation deficient mutant GFP-NRAS[SAAX] was used as a negative control for this experiment, as unprenylated NRAS is not a substrate for ICMT.

We expressed GFP-tagged HRAS, NRAS, KRAS4A, or KRAS4B in these cells and analyzed subcellular localizations by live-cell confocal imaging. As a control we expressed GFP-RIT, which contains a non-CaaX PM targeting C-terminus (Heo et al, 2006). As expected, in sg[Tom]-targeted cells all GFP-tagged RAS proteins and GFP-RIT decorated the PM. GFP-NRAS was also observed on a paranuclear structure that we previously showed to be the Golgi apparatus (Choy et al, 1999). The localizations of GFP-tagged HRAS, KRAS4A, KRAS4B, and RIT were unchanged in sg[ICMT]-targeted cells (Fig 1D). In contrast, GFP-NRAS was dramatically mislocalized to the cytosol with complete removal from the PM and enhancement of Golgi decoration (Fig 1D). The unique sensitivity of NRAS to mislocalization upon disrupting *ICMT* was surprising. We also confirmed that GFP-NRAS was mislocalized from the PM into the cytosol in U2OS cells in which *ICMT* was targeted by CRISPR/Cas9 as well as with the complementary methods of RNAi and pharmacological inhibition with CMPD75 (Judd et al, 2011) (Fig 1E). This finding also demonstrates that the requirement of ICMT for the correct trafficking of GFP-NRAS to the PM seems independent of the cell of origin, as similar results were observed in melanoma cells and U2OS, an osteosarcoma cell line, driven by mutations in NF1 and TP53. Importantly, the PM localization of GFP-NRAS could be restored by ectopic expression of ICMT (Fig 1F). Thus, ICMT activity is required for PM targeting of NRAS but not other RAS isoforms.

The subcellular localization of endogenous RAS proteins cannot be determined by immunofluorescence because none of the many anti-RAS antibodies are sufficiently sensitive (Waters et al, 2017). We have therefore studied the localization of endogenous RAS proteins by subcellular fractionation using nitrogen cavitation followed by differential centrifugation (Choy et al, 1999; Zhou and Philips, 2017). We previously reported that NRAS was unique among the four RAS proteins in that the bulk of the endogenous protein is found not associated with membranes but rather in the cytosol in a de-palmitoylated state associated with chaperones that include VPS35 (Zhou et al, 2016) and PDE6$\delta$ (Chandra et al, 2011). Consistent with these studies, 58% of endogenous NRAS in SKMEL28 cells was recovered in the S100 cytosolic

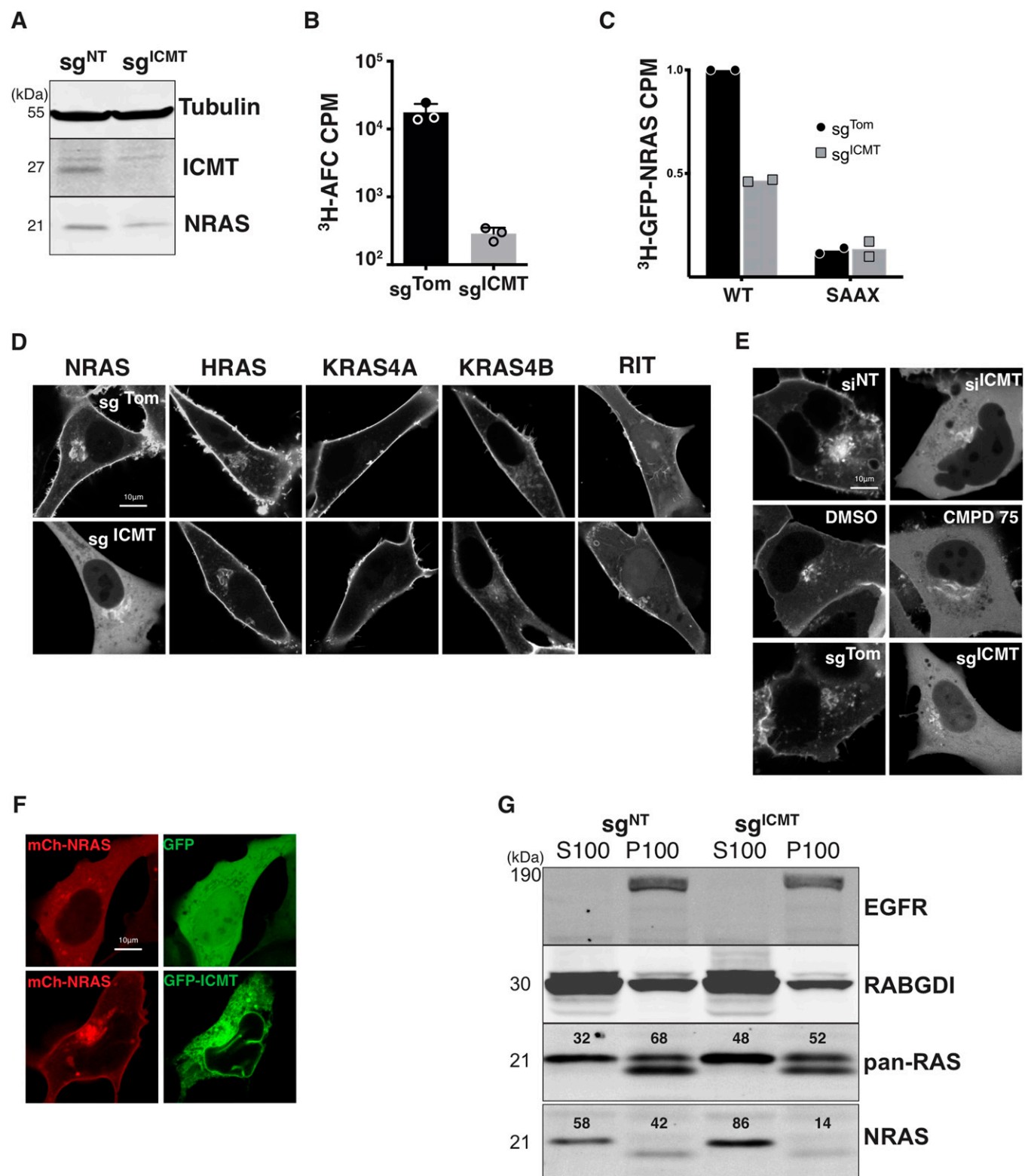

**Figure 1. ICMT is required for the plasma membrane expression of NRAS, but not other RAS isoforms.**
**(A)** Lysates from SKMEL28 cells ± sg[ICMT] were immunoblotted using the indicated antibodies. **(B)** The prenylcysteine-directed methylation activity present in isolated membranes of SKMEL28 cells ± sg[ICMT] was assessed in vitro using [$^3$H]-methyl-adenosyl-l-methionine and $N$-acetyl-$S$-farnesyl-l-cysteine as a methyl donor and acceptor, respectively. **(C)** Carboxyl methylation of GFP-NRAS and the prenylation deficient mutant GFP-NRAS[SAAX] in vivo were assessed in SKMEL28 cells ± sg[ICMT] via metabolic labelling with l-[$methyl$-$^3$H]methionine. **(D, E)** The steady-state distributions of GFP-tagged full length RAS or RIT small G-proteins were visualized by live-cell fluorescence confocal microscopy in SKMEL28 cells ± sg[ICMT] (E). **(F)** GFP-NRAS was visualized in U2OS cells in which ICMT was inhibited via siRNA, a small molecule ICMT

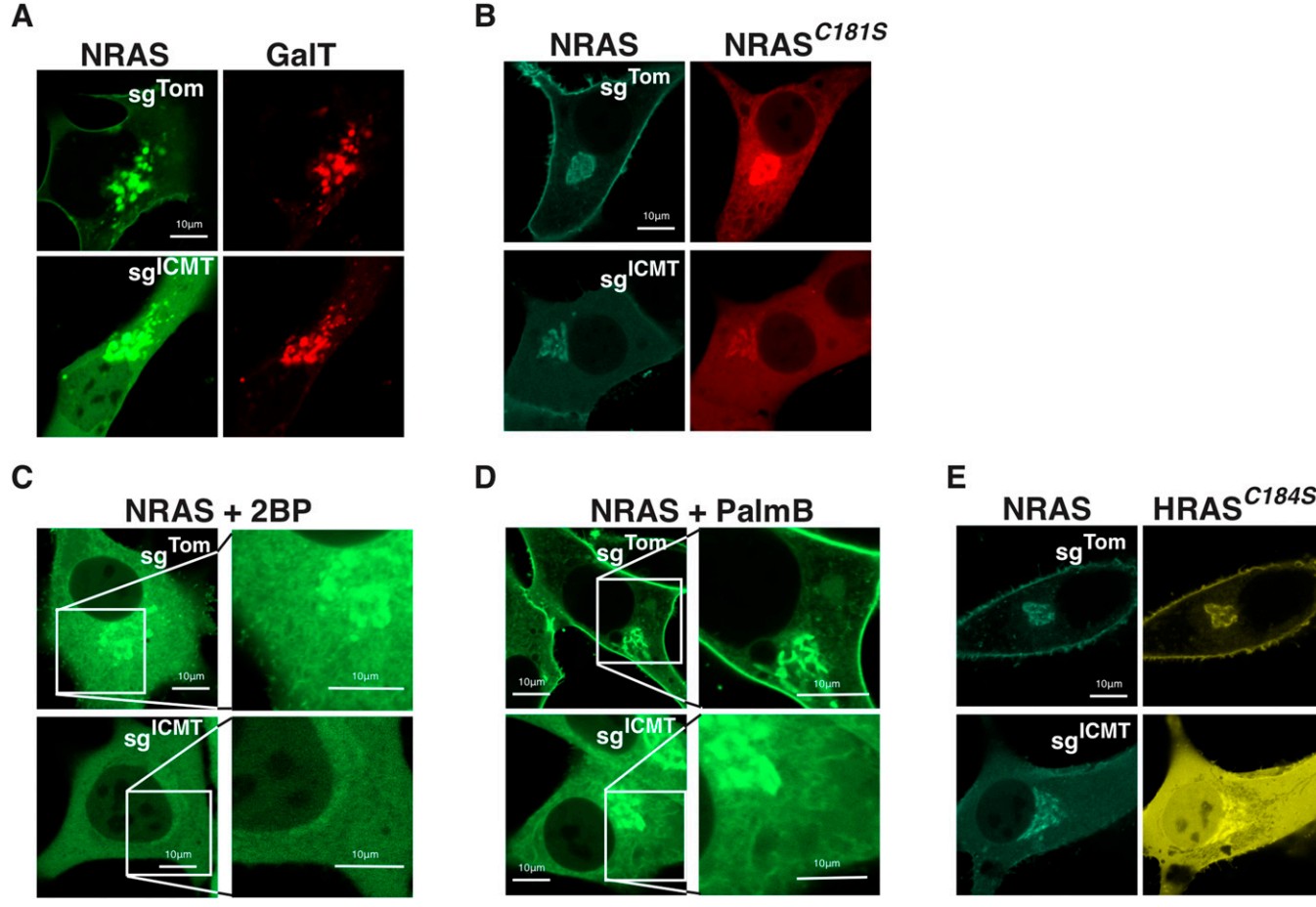

**Figure 2.  Carboxyl methylation and palmitoylation regulate NRAS endomembrane trafficking distinctly.**
**(A)** Co-expression of GFP-NRAS and mKO-GalT, a marker for the golgi, was examined by live-cell fluorescence confocal microscopy in U2OS cells ± sg[ICMT].
**(B)** mTurquoise-NRAS co-expressed with palmitoylation-deficient mCherry-NRAS[C181S] in SKMEL28 cells ± sg[ICMT] and imaged as above. **(C, D)** GFP-NRAS expressed in SKMEL28 cells ± sg[ICMT] treated with either the palmitoylation inhibitor 2-bromopalmitate, 2BP, or instead with palmostatin B, PalmB (D), an inhibitor of enzymatic depalmitoylation. Insets are magnified views to demonstrate endomembranes morphologically consistent with the golgi and ER. **(E)** mTurquoise-NRAS co-expressed with a single site palmitoylated mutant YFP-HRAS[C184S] in SKMEL28 cells ± sg[ICMT] and imaged as above. Bars indicate 10 μm.

fraction (Fig 1G). Disrupting *ICMT* in these cells increased to 86% the fraction of endogenous NRAS recovered in the S100, demonstrating that the results described above for GFP-NRAS reflected the localization of the endogenous protein.

### ICMT loss reduces NRAS membrane affinity but golgi membrane binding is preserved

To confirm that the persistent paranuclear fluorescence of GFP-NRAS observed in the absence of ICMT represents decoration of Golgi membranes, we co-expressed CFP-NRAS and YFP-GalT and observed strong co-localization (Fig 2A). GalT is galactyltransferase expressed in the Golgi, and is used as a marker for Golgi localization. Thus, unlike

association with the PM, NRAS association with Golgi membranes does not require carboxyl methylation. NRAS and HRAS are palmitoylated on the Golgi by the protein acyltransferase (PAT) DHHC9/GCP16 (Swarthout et al, 2005). Unlike prenylation, both carboxyl methylation and palmitoylation of GTPases are reversible. We thus sought to determine if these modifications were interdependent. We have previously reported that NRAS C181S that cannot be palmitoylated accumulates on the Golgi, ER, and nuclear envelope and does not traffic to the PM (Choy et al, 1999). We confirmed this localization in SKMEL28 cells (Fig 2B). In the setting of ICMT deficiency, non-palmitoylated GFP-NRAS[C181S] was predominantly cytosolic and did not decorate ER but retained affinity for the Golgi (Fig 2B). Similar patterns were observed when palmitoylation was inhibited by 2-bromopalmitate (Fig 2C). Thus, ICMT is required for

---

inhibitor CMPD75, or by *ICMT*-targeted genomic editing (F). Ectopic expression of GFP-ICMT in sg[ICMT] targeted SKMEL28 cell rescues proper NRAS trafficking.
**(G)** Immunoblotting using the indicated antibodies was used to assess partitioning of endogenous NRAS in SKMEL28 cells ± sg[ICMT] between cytosolic (S100) and membrane (P100) fractions recovered after cellular disruption via nitrogen cavitation and fractionation by ultracentrifugation. EGFR and RABGDI serve as markers of membrane and cytosol fractions, respectively. Results shown are representative of two independent experiments. Bars indicate 10 μm. CPM, counts per minute.
Source data are available for this figure.

association of non-palmitoylated NRAS with the ER but not the Golgi and both modifications are required for trafficking to the PM. Palmostatin B (PalmB) has been shown to inhibit depalmitoylation of NRAS by ABHD17 thioesterases (Lin & Conibear, 2015). Treatment of SKMEL28 cells with PalmB enhanced localization of GFP-NRAS on Golgi and PM (Fig 2D). In ICMT-deficient cells, however, PalmB was unable to reestablish expression of GFP-NRAS on the PM and the protein remained predominantly localized in the cytosol and on the Golgi (Fig 2D). Together these data suggest that palmitoylation/depalmitoylation cycling and carboxyl methylation function cooperatively to support NRAS trafficking to the PM. Unexpectedly, association of NRAS with the Golgi required neither palmitoylation nor carboxyl methylation, although association with ER membranes in the absence of palmitoylation did require carboxyl methylation.

Whereas NRAS is modified by a single palmitate on cysteine 181, HRAS is modified with two palmitoyl acyl chains at cysteines 181 and 184 (Hancock et al, 1989). The differential sensitivity of NRAS and HRAS to mislocalization (Fig 1) in the setting of ICMT-deficiency suggested that the number of palmitate modifications dictates sensitivity. To test this hypothesis, we mutated cysteine 184 to serine in HRAS to remove one of the palmitoylation sites. Singly palmitoylated HRAS[C184S] colocalized precisely with NRAS and did not reach the PM in ICMT-deficient cells (Fig 2E). This suggests that dual acylation affords sufficient affinity for membranes such that carboxyl methylation of the prenylcysteine is not required. These data also demonstrate that carboxyl methylation is not a prerequisite for palmitoylation.

### Carboxyl methylation promotes NRAS palmitoylation

To more directly examine whether palmitoylation of NRAS depends on carboxyl methylation we measured palmitoylation in the presence or absence of ICMT. We used acyl-resin assist capture to biochemically enrich for endogenous palmitoylated proteins and used immunoblotting to identify palmitoylated NRAS in SKMEL147 cells ± ICMT inhibition (Fig 3A). As predicted, 2BP treatment of cells before protein purification eliminated the pool of palmitoylated NRAS that was captured by thiopropyl sepharose beads. The pool of palmitoylated NRAS detected by this method was affected neither by disrupting ICMT with CRISPR/cas9 nor by treatment with CMPD75. In contrast, both genetic ablation and pharmacologic inhibition of ICMT diminished the abundance of [3]H-palmitate incorporated into GFP-NRAS upon metabolic labelling (Fig 3B). Thus, although ICMT is not required for palmitoylation, NRAS is more efficiently palmitoylated when carboxyl methylated.

### Carboxyl methylation and palmitoylation regulate NRAS golgi trafficking

NRAS cycling between the Golgi and PM is driven by a palmitoylation/depalmitoylation cycle (Rocks et al, 2005). As such, the reduced palmitoylation of NRAS when ICMT was inhibited suggested that proper acylation cycling requires ICMT. To test this hypothesis, we performed kinetic analyses of GFP-NRAS trafficking. To determine the effect of ICMT deficiency on anterograde trafficking of NRAS away from the Golgi we used photoactivatable GFP (paGFP) (Fig 4A). Once irreversibly activated, paGFP-NRAS was lost from the Golgi much faster in ICMT-deficient cells than in ICMT replete cells (Fig 4B). This result suggests that, although unmethylated NRAS can associate with the Golgi, it has a reduced affinity for Golgi membranes, leading to a shorter dwell time on this compartment, which enhances anterograde flux. To measure retrograde transfer of NRAS from the PM and cytosol to the Golgi we used FRAP (Fig 4C). We measured the recovery of GFP-NRAS on the Golgi after photobleaching in the presence or absence of ICMT and found that recovery was accelerated in the absence of ICMT (Fig 4D, left). The same result was obtained when ICMT

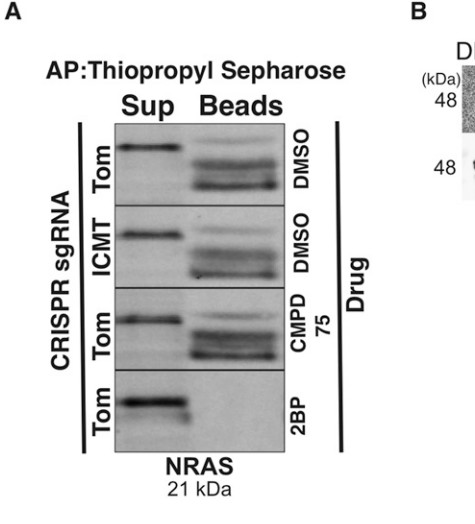

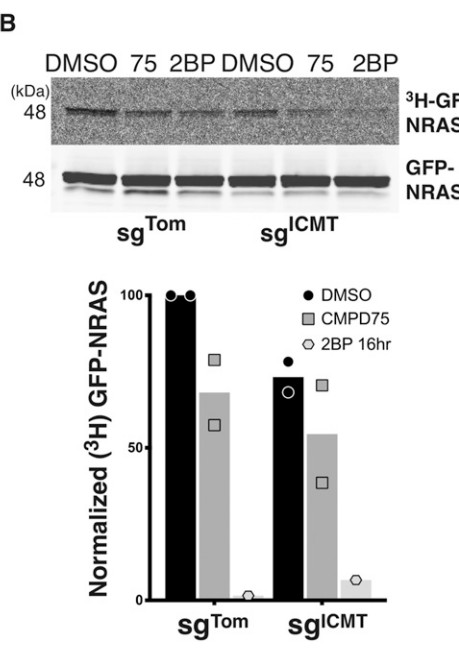

**Figure 3. ICMT inhibition reduces NRAS palmitoylation.**
**(A)** Endogenous palmitoylated NRAS was affinity purified via acyl-resin assist capture from SKMEL147 cells ± sg[ICMT] in the presence or absence of CMPD75 or 2-BP, followed by detection via immunoblot. **(B)** Palmitoylation of GFP-NRAS was quantified by metabolic labelling of SKMEL147 cells ± sgICMT with [3]H-palmitic acid in the presence or absence of CMPD75 or 2BP, followed by immunoprecipitation, SDS–PAGE/Western, and then by autoradiography. Graph represents normalized ratio of the autoradiography signal intensity measured with ImageJ to the GFP Western blot signal intensity quantified via LiCor image analysis from two independent experiments. Source data are available for this figure.

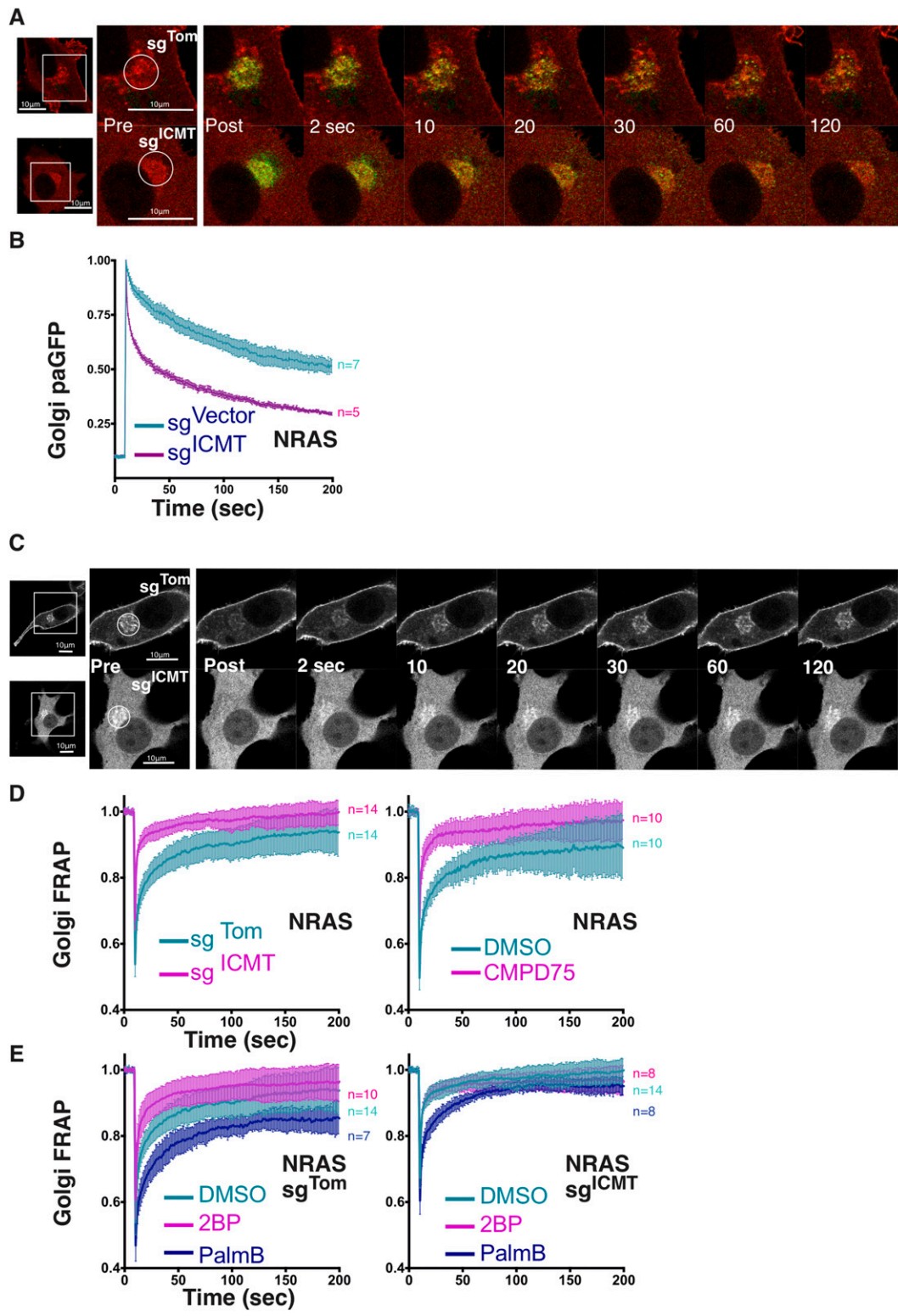

**Figure 4.   ICMT inhibition accelerates GFP-NRAS golgi trafficking flux.**
**(A)** Anterograde trafficking of NRAS away from the Golgi was assessed via photoactivation of paGFP-NRAS on the Golgi (marked by mCherry-NRAS) and monitoring fluorescence loss over time; a representative cell is shown. **(B)** Curves trend the loss of initial fluorescence post-photoactivation in the region of the Golgi in cells ± sgICMT as mean ± SEM for cells examined (n = indicated). **(C)** Retrograde trafficking of NRAS to the Golgi was examined using GFP-NRAS expressed in SKMEL28 cells with FRAP kinetics determined in the region of the Golgi; a representative cell is shown. **(D, E)** Recovery curves trend the mean ± SEM of the calculated fraction of initial fluorescence over time in the cells examined (n = indicated). Experimental conditions include ± sgICMT (D, left panel), Cmpd75 (D, right panel), or ± sgICMT in the presence or absence of 2-BP or PalmB (E, left and right). Bars indicate 10 $\mu$m.

was inhibited with CMPD75 (Fig 4D, right). Accelerated recovery indicates either higher efficiency of capture or greater retrograde flux. Because the photoactivation study of anterograde trafficking is inconsistent with higher affinity for Golgi membranes we conclude that the accelerated recovery reflects increased flux from the donor compartments. Consistent with this interpretation, inhibiting palmitoylation with 2-BP also accelerated recovery, presumably because of the increased cytosolic donor pool (Fig 4E). Interestingly, the effect of 2-BP was lost in cells deficient in ICMT suggesting that lack of palmitoylation could not increase the cytosolic donor pool more than that accomplished by loss of carboxyl methylation. By blocking depalmitoylation, PalmB should stabilize NRAS at membranes, retard retrograde trafficking of NRAS from the PM to the Golgi, and thereby slow Golgi FRAP kinetics. This was indeed the case (Fig 4E). In fact, whereas 2-BP did not change Golgi FRAP kinetics of unmethylated GFP-NRAS, PalmB treatment did retard Golgi FRAP kinetics even when ICMT was inhibited. This result confirmed that the acylation cycling of NRAS continues in the absence of ICMT and further suggests that depalmitoylation may be accelerated in the absence of carboxyl methylation, perhaps contributing to the observed diminished palmitate incorporation (Fig 3B).

### Carboxyl methylation regulates PDE6δ-mediated NRAS Golgi trafficking

To determine the role of carboxyl methylation in association of GFP-NRAS with the cytosolic chaperone PDE6δ, we studied the effect of overexpressing or silencing PDE6δ on the subcellular distribution of GFP-NRAS in the presence or absence of ICMT. As we previously reported in HEK293 cells (Tsai et al, 2015), overexpressing PDE6δ in U2OS cells resulted in a redistribution of GFP-NRAS to the cytosol, demonstrating the ability of the chaperone to extract NRAS from membranes and participate in trans-cytosolic trafficking (Fig 5A). Silencing PDE6δ with siRNA did not affect association of GFP-NRAS with the PM but resulted in enhanced expression on the Golgi in SKMEL173 cells (Fig 5B), indicating that removal of the GTPase from this compartment (anterograde trafficking) is promoted by PDE6δ. Silencing PDE6δ in conjunction with ICMT did not prevent accumulation of GFP-NRAS in the cytosol (Fig 5B). This result indicates that PDE6δ is not required for cytosolic sequestration of prenylated NRAS and raises the possibility that other prenyl-binding chaperones may recognize unmethylated NRAS in the cytosol. Deltarasin is a drug that prevents the binding of RAS to PDE6δ (Zimmermann et al, 2013). Similar to silencing PDE6δ, treatment of cells with deltarasin enhanced the accumulation of GFP-NRAS on Golgi membranes, consistent with a function for PDE6δ in promoting anterograde trafficking from the Golgi. When ICMT was inhibited by with siRNA or with CMPD75, neither PDE6δ siRNA nor deltarasin treatment could reestablish PM localization. These findings suggested that mislocalization of unmethylated GFP-NRAS was not a consequence of disrupted PDE6δ-mediated trafficking.

To directly assess a role for ICMT in PDE6δ-dependent NRAS trafficking we again used FRAP and photoactivation of GFP. Neither overexpression of PDE6δ nor inhibition of endogenous PDE6δ with deltarasin affected FRAP (Fig 5C). Similar results were obtained in the absence of ICMT (Fig 5C) suggesting that retrograde trafficking does not require PDE6δ. In contrast, anterograde trafficking measured by loss of photoactivated paGFP-NRAS from the Golgi was markedly accelerated by overexpression of PDE6δ (Fig 5D). This

accelerated trafficking was not apparent when ICMT was inhibited, however, indicating that PDE6δ acts to facilitate post-Golgi transport only of methylated GFP-NRAS. Our observation that in the absence of PDE6δ overexpression ICMT deficiency accelerates loss of paGFP-NRAS from the Golgi (Fig 4B) suggests that efflux from the Golgi may take both a PDE6δ dependent and independent path that differentially require carboxyl methylation.

To determine directly the effect of carboxyl methylation on the binding of NRAS to PDE6δ we measured the association using FLIM-FRET (Fig 5E and F). Deltarasin served as the positive control and reduced the FRET efficiency by 50%. Strikingly, silencing *ICMT* also significantly reduced FRET efficiency. This result demonstrates that PDE6δ binds NRAS more efficiently when it is methylated, concordant with previous reports of binding to KRAS4B (Dharmaiah et al, 2016). Importantly, this result also suggests that the cytosolic accumulation of unmethylated NRAS, its rapid flux both on and off the Golgi, and its restriction from the PM is not the consequence of enhanced sequestration by PDE6δ.

## Discussion

We have shown that efficient trafficking of NRAS, but not other RAS isoforms, to the PM requires ICMT-mediated carboxyl methylation. Methylation is more critical for NRAS trafficking than other RAS proteins because of the labile and comparably weak contribution that single site, reversible palmitoylation contributes to membrane affinity compared with either dual acylation or the electrostatic affinity afforded by polybasic domain(s), which are present in other RAS proteins. Reduced palmitoylation in the absence of ICMT likely indicates greater lability of the palmitate modification on NRAS, and this hypothesis is supported by the observation that an inhibitor of depalmitoylation, PalmB, was able to retard FRAP kinetics of GFP-NRAS at the Golgi when ICMT was inhibited. However, PalmB treatment was still unable to establish PM localization of unmethylated NRAS. Although unmethylated NRAS was able to associate with Golgi membranes and to become palmitoylated, flux to and from this organelle was markedly accelerated. This altered flux may restrict the capacity of NRAS to traffic to post-Golgi compartments before trafficking to the PM, such as the recycling endosomal compartment (Misaki et al, 2010). We conclude that in the absence of ICMT, NRAS has reduced affinity for the Golgi resulting in a shorter dwell time on this compartment, which likely precludes efficient sorting onto transport vesicles destined for the PM. Inhibition of palmitoylation has been suggested as a therapeutic modality for NRAS-driven malignancies (Xu et al, 2012; Zambetti et al, 2020). Our results suggest that ICMT inhibitors alone or in combination with palmitoylation inhibitors may also have NRAS-specific therapeutic potential.

## Materials and Methods

### Plasmids, cell culture, and siRNA

LentiCRISPR v2 was obtained courtesy of F Zhang; Massachusetts Institute of Technology; #52961; Addgene. pEGFP-C3 vector was obtained from Takara Bio Inc. pEGFP-HRAS, EGFP-NRAS, EGFP-KRAS4A,

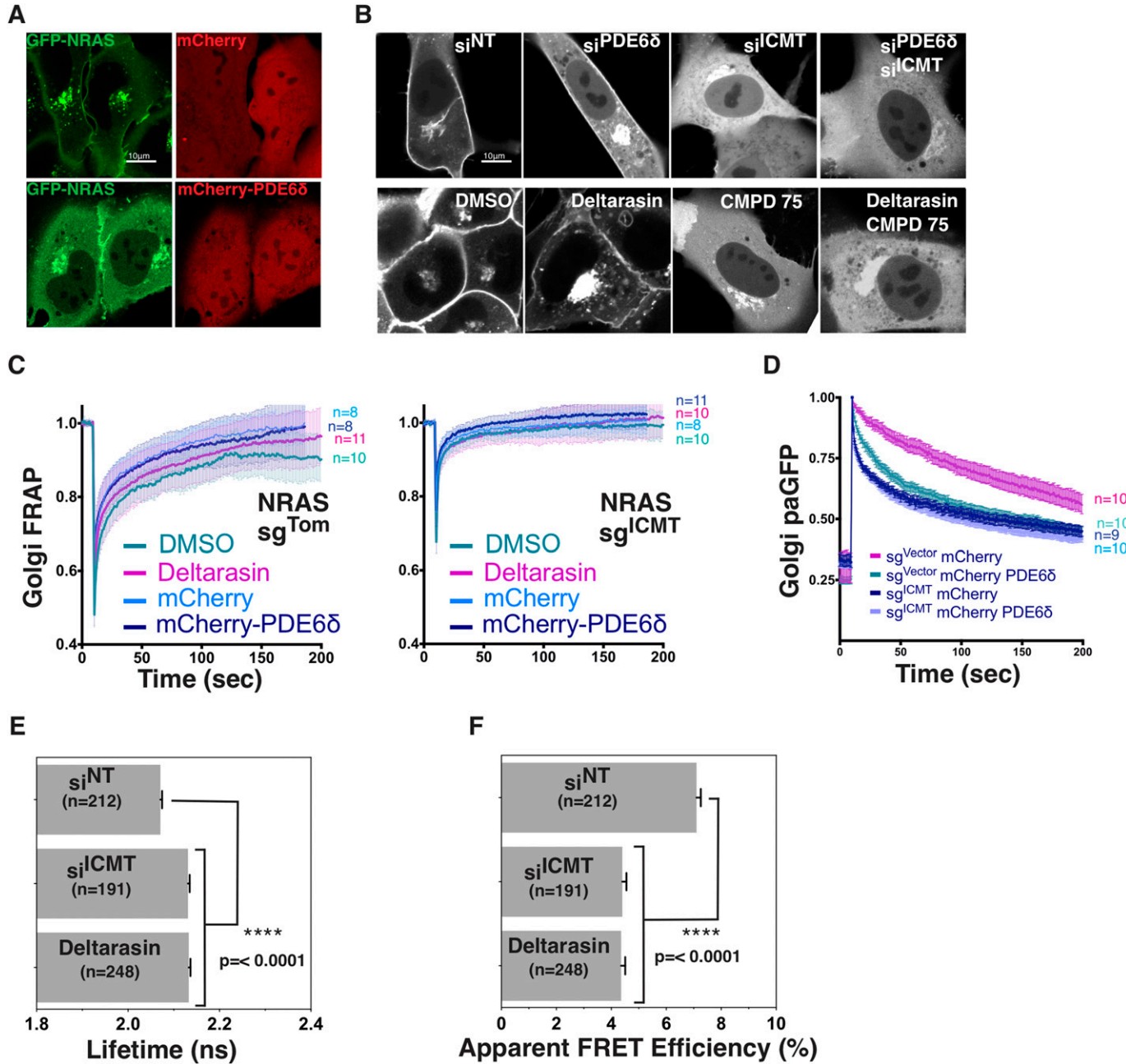

**Figure 5. ICMT promotes NRAS-PDE6δ interactions and anterograde NRAS Golgi trafficking, NRAS plasma membrane trafficking does not require PDE6δ expression and does require ICMT.**
**(A)** Confocal fluorescence micrographs of GFP-NRAS co-expressed with mCherry or mCherry-PDE6δ in U2OS cells. **(B)** Confocal fluorescence micrographs of GFP-NRAS expressed in SKMEL173 cells treated with control si[cont], si[PDE6δ], si[ICMT], or si[PDE6δ] with si[ICMT] (upper panel) or with DMSO, deltarasin (PDE6δ inhibitor), CMPD75, or deltarasin and CMPD75 (lower panel). **(C, D)** Golgi FRAP for GFP-NRAS and loss of Golgi paGFP-NRAS (D) kinetics in SKMEL28 cells ± sgICMT with co-expression of either mCherry or mCherry-PDE6δ. Bars indicate 10 μm. **(E, F)** N-RasG12V/PDE6δ interaction-FLIM-FRET. HEK293 EBNA cells were co-transfected with pmGFP-tagged N-RasG12V and pmCherry tagged PDE6δ. As indicated cells were concurrently transfected with siRNA against ICMT or scrambled siRNA in 0.1% DMSO (control). Deltarasin treatment (2.5 μM) was for 24 h after transfection. Statistical significance of differences between control and treated cells were examined using one-way ANOVA complemented with Tukey's test (ns, not significant; **$P < 0.01$, ***$P < 0.001$, ****$P < 0.0001$).

EGFP-KRAS4B, EGFP-RIT, ECFP-NRAS were engineered from base vectors (Clontech) and mKO-GALT was obtained courtesy of M Davidson; Florida State University; #57843; Addgene. HEK293 cells were obtained from American Type Culture Collection. SKMEL28, SKMEL173, and SKMEL147 cells were gifts from E Hernando (New York University Langone Medical Center). Our U2OS cells were obtained from American Type Culture Collection. All cells were maintained in DMEM containing 10% FBS and 1% penicillin/streptomycin at 37°C and 5% $CO_2$. The following compounds were also used during cell culture at the indicated concentrations: Compound 75 (gift from M

Bergö); 2-bromopalmitate (Sigma-Aldrich); Palmostatin B (Calbiochem). Knockdown using siRNA (excepting FLIM-FRET, see below) was carried out by transfecting cells with ICMT, PDE6δ, or non-targeting SMARTpools siRNAs (Dharmacon, Life Technologies) using Dharmafect 1 (Dharmacon, Life Technologies) transfection reagent according to the manufacturer's instructions.

## Confocal microscopy

Live-cell confocal imaging was performed using an LSM 800 inverted confocal microscope running Zen Blue software (ZEISS). Live-cell imaging was performed at 37°C and 5% $CO_2$ in DMEM/10% FBS media. All cells were imaged with a 63× 1.4 NA objective. Cells were transfected with plasmids containing the required fluorescent protein-tagged construct using Lipofectamine 3000 (Invitrogen, Life Technologies) 1 d before imaging.

## CRISPR/Cas9

Genomic disruption of *ICMT* was performed by infecting SKMEL28, SKMEL147 cells with lentivirus generated by transfecting HEK293 cells with pLentiCRISPR v2 with a sgRNA targeting exon 1 of *ICMT* (sg[ICMT], 5'-CACCGCACCGGGCTGGCGCTCTACG-3' and 5'-AAACCGTAGA GCGCCAGCCCGGTGC-3'), and Cas9 using Lipofectamine 3000 (Invitrogen, Life Technologies). Control cells were made by infecting cells with lentivirus generated by transfecting HEK293 cells with pLentiCRISPR v2 with a sgRNA targeting Tomato fluorescent protein (sg[Tom], 5'GCCACGAGTTCGAGATCGA and 5'-TCGATCTCGAACTCGTGGC) and Cas9. Cells were selected with 2 µg/ml puromycin 2 d after infection and used immediately for experiments.

## Subcellular fractionation

Membrane (P100) and cytosolic (S100) fractions were isolated from SKMEL28 cells as previously described (Zhou and Philips, 2017). In brief, adherent SKMEL28 cells were washed with cold PBS and collected by scraping off tissue culture plates and centrifugation at 500*g*. Cell pellets were resuspended in ice-cold hypotonic buffer (100 mM KC1, 3 mM NaCl, 3.5 mM $MgCl_2$, 10 mM Hepes, pH 7.3, and protease inhibitors) and spun in a bomb (Parr Instruments) pressurized with nitrogen to 450 ψ for 20 min on ice. The pressurized cell suspensions were released dropwise to room atmospheric pressure and collected in tubes, then centrifuged at 20,000*g* using an Eppendorf 5427R rotor and centrifuge, for 10 min at 4°C to remove nuclei and unbroken cells. The post-nuclear supernatant was then centrifuged with a TLA100.3 rotor (Beckman Coulter) at 100,000 rpm (350,000*g*) for 30 min. The supernatant (S100 cytosolic fraction) was carefully removed, and the pellet (P100 membrane fraction) was washed twice with cold hypotonic buffer and resuspended in 25 mM Tris, pH 7.4, plus protease inhibitors to one-tenth the volume of the S100 fraction. For SDS–PAGE, the S100 and P100 fractions were mixed with 4× Laemmli sample buffer (Bio-Rad Laboratories) and 20 mM DTT (Sigma-Aldrich) and then were loaded in a 10:1 ratio (S100/P100) to maintain cell-equivalent quantities. For ICMT activity assays, the membrane protein concentration was quantified using a BCA protein assay (Thermo Fisher Scientific).

## In vitro and in cell carboxyl methylation assays

An in vitro assay for ICMT activity was performed using 10 µg of protein from isolated membrane fractions from SKMEL28 cells previously infected with either lentivirus expressing sg[ICMT]/Cas9 or sg[Tom]/Cas9 as described previously (Choy & Philips, 2000). Briefly, 10 µg of membrane protein was incubated for 30 min at 37°C with 5 µl of 1 mM *N*-acetyl-*S*-farnesyl-l-cysteine (AFC; assay substrate), 3 µl adenosyl-l-*S*-[³H]methionine ([³H]AdoMet; 60 Ci/mmol, 0.55 mCi/ ml; methyl donor), and 12.5 µl 4× TE buffer (200 mM Tris–HCl, pH 8.0, and 4 mM Na-EDTA) in a volume of 50 µl. An equal vol of 20% TCA was then added to terminate the reaction. Next, 400 µl *n*-heptane was added to separate the unreacted [³H]AdoMet from the [³H] AFC–methyl ester product in the top organic layer. The top *n*-heptane layer was removed after centrifugation and evaporated overnight. 1 N NaOH was used to hydrolyze the [³H]AFC-methyl ester to produce volatile [³H] methanol, which was measured using a scintillation counter. An in-cell assay for ICMT activity in the same cells was performed as described previously (Choy & Philips, 2000). Briefly sg[ICMT] and sg[Tom] cells were transfected with a plasmid containing GFP-NRAS using Lipofectamine 3000 (Thermo Fisher Scientific). 1 d later, the cells were washed in PBS and incubated in DMEM without methionine/cysteine with 10% dialyzed FBS for 3 h. 200 µCi of [³H methyl]–methionine was then added to the media and the cells incubated for 3 h, washed in PBS, and then lysed with RIPA buffer (20 mM Tris-HC1, pH 7.5, 150 mM NaCl, 1% NP-40, 0.1% SDS, 0.1% Na-deoxycholate, 0.5 mM EDTA, 1 mM DTT, and protease inhibitors). GFP-NRAS was then immunoprecipitated using anti-GFP agarose beads (MBL International), the beads were washed in RIPA buffer and subjected to SDS–PAGE. The gel was dried (DryEase Mini Gel Drying System; Novex), and the areas of the gel containing GFP-NRAS were cut out and treated with 1 N NaOH in open-topped microcentrifuge tubes to produce to produce volatile [³H] methanol via alkaline hydrolysis of the [³H] methyl ester then carefully sealed in a prepared scintillation vials for 3 d and volatilized ³H was measured by scintillation counting.

## Immunoblot

Cells were lysed in Tris-glycine SDS sample buffer (Thermo Fisher Scientific) with 20 mM DTT (Sigma-Aldrich), and the lysates were subjected to SDS–PAGE and Western blotting on polyvinylidene flouride (PVDF) membranes (Bio-Rad Laboratories). Membranes were blocked (Odyssey blocking buffer; LI-COR Biosciences) and incubated with the following primary antibodies: anti-Tubulin (clone E7-s; DSHB), anti- ICMT (Cat. no. 51001-2-AP; ProteinTech Group), anti-NRAS (sc-31, F155; Santa Cruz), anti-EGFR (sc-120; Santa Cruz), anti-RABGDI (sc-374649; Santa Cruz), anti-pan RAS (Cat. no. OP40, RAS10; Calbiochem), anti-GFP (Ref A-6455; Thermo Fisher Scientific). Secondary antibodies used were IRDye 800–conjugated goat anti–rabbit or IRDye 680–conjugated goat anti–mouse (926-32211 and 926-68070; LI-COR Biosciences). Blots were visualized with the Odyssey infrared imaging system (LI-COR Biosciences) and quantified using Odyssey software.

Endogenous NRAS palmitoylation was detected using the CAP-TUREome S-Palmitoylated Protein Kit (Badrilla) according to the manufacturer's protocol. Briefly, 6 × 10⁶ SKMEL147 cells were cultured

overnight in the presence of DMSO, 5 µM CMPD75, or 50 µM 2-BP and used as starting material. Proteins affinity purified on thiopropyl sepharose beads were analyzed by SDS–PAGE, transferred to PVDF, immunoblotted with anti-NRAS mAb (sc-31, F155; Santa Cruz), and detected via LiCor infrared scanning. Quantitative radiometric detection of GFP-NRAS palmitoylation was performed via overnight metabolic labelling with $^3$H-palmitic acid (Perkin Elmer) as previously described (Tsai and Ahearn). GFP proteins were immunoprecipitated size fractionated by SDS–PAGE, transferred to PDVF, and Western blotted with anti-GFP mAb before detection via LiCor infrared scanning. Autoradiographic detection of $^3$H signal was performed by exposing blotted PVDF membranes to Kodak BioMax MS film (CareStream) in a BioMax Transcreen-LE Intensifying Screen (Care-Stream) for 2 d. Film was developed, scanned, and band densitometry was measured using ImageJ and used to normalize to GFP protein expression as determined via LiCor quantitation.

### FRAP and photoactivation

FRAP was performed on genome edited or overnight drug treated SKMEL28 cells transfected with GFP-NRAS. Zeiss Zen Blue software operating a Zeiss LSM 800 laser scanning confocal microscope fitted with a Plan-Apochromat 63× NA 1.4 oil immersion lens was used to analyze live cells at 37°C and 5% $CO_2$ in 10% FBS containing DMEM media. Circular 9 µM regions of interest (ROI) were drawn around the regions of the Golgi, which were subsequently bleached using 100% 488 nm laser power for five iterations after 10 initial scans. Serial 512 × 512 pixel resolution micrographs were collected using a minimal pixel dwell time and a maximum scan speed every 0.5 s for 2 min. Integrated ROI fluorescence intensities were corrected for background fluorescence, normalized to total cell fluorescence at each time point to account for scan-related bleaching, and the fluorescence recovery was graphed as the fraction of initial fluorescence. Curves are constructed from mean calculated recovery values ± SEM for the number of cells analyzed, as indicated. All data were acquired using identical bleach and acquisition settings for all cells and conditions tested. Kinetic tracing of photoactivated GFP-NRAS (paGFP-NRAS) was performed analogously by photoactivating 9-µm circular ROI encompassing the Golgi apparatus in cells co-expressing pa-GFP-NRAS and mCherry-NRAS using a 405 nm laser at 100% power for five iterations followed by serial scanning every 0.5 s, as noted above. Kinetic analyses from acquired fluorescence intensities again included correction for background and normalization total cell fluorescence. Data points reflect the mean fluorescence intensity ± SEM over time relative to the maximum fluorescence generated immediately post-activation.

### FLIM FRET

80,000 HEK293 EBNA cells were seeded in 12 well plates onto sterile cover slips. The next day cells were transfected either with pmGFP-NRasG12V only (donor control) or with FRET pair plasmids pmGFP-NRasG12V and pmCherry-PDEδ at a ratio of 1:3 (1 µg plasmids total). JetPRIME transfection reagent (Polyplus transfection) was used according to the manufacturer's instructions. ICMT or non-targeting SMARTpool siRNAs (Dharmafect, Life Technologies) were transfected together with the FRET pair plasmids. Alternatively, cells were treated with 2.5 µM deltarasin 24 h after transfection. Scrambled siRNA in 0.1% DMSO served as control. After 24 h of treatment cells

were fixed with 4% PFA for 12 min and mounted with Mowiol 4-88 (#81381; Sigma-Aldrich). The lifetime of the donor mGFP was measured using a fluorescence microscope (Zeiss AXIO Observer D1) with fluorescence lifetime imaging attachment from Lambert Instruments as previously described in (Guzmán et al, 2016; Posada et al, 2017). At least 50 cells were selected to measure the lifetime in each sample. The apparent FRET efficiency percentage (Eapp) was calculated using the equation Eapp = (1 − $\tau_{DA}/\tau_D$) × 100%, where $\tau_{DA}$ is the lifetime of the donor in the presence of acceptor (FRET-sample) and $\tau_D$ is the lifetime of the donor only.

## Supplementary Information

## Acknowledgements

We thank Dr. Eva Hernando for cell lines used in this work and Dr. Martin Bergö for compound 75. This project was supported by a Dermatology Foundation Research Grant and T32AR064184 funding to IM Ahearn and NIH R35CA253178 to MR Philips. D Abankwa acknowledges support from the Academy of Finland (#304638) and the Jane and Aatos Erkko Foundation, Finland. F Siddiqui acknowledges support from the Finnish National Agency for Education and Åbo Akademi University.

### Author Contributions

IM Ahearn: conceptualization, formal analysis, supervision, funding acquisition, investigation, methodology, project administration, and writing—original draft, review, and editing.
HR Court: conceptualization, formal analysis, investigation, and writing—original draft, review, and editing.
F Siddiqui: formal analysis, investigation, methodology, and writing—original draft.
D Abankwa: formal analysis, investigation, methodology, and writing—original draft.
MR Philips: conceptualization, formal analysis, supervision, funding acquisition, investigation, methodology, project administration, and writing—original draft, review, and editing.

### Conflict of Interest Statement

The authors declare that they have no conflict of interest.

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
