## [Reviewer comments · Life Science Alliance]

Life Science Alliance

NRAS is Unique Among RAS Proteins in Requiring ICMT for Trafficking to the Plasma Membrane

Ian Ahearn, Helen Court, Farid Siddiqui, Daniel Abankwa, and Mark R. Philips

DOI: <https://doi.org/10.26508/lsa.202000972>

Corresponding author(s): Ian Ahearn, NYU Langone Medical Center and NYU Grossman School of Medicine

Review Timeline:

Submission Date:	2020-11-24
Editorial Decision:	2021-01-13
Revision Received:	2021-01-26
Accepted:	2021-01-27

Scientific Editor: Shachi Bhatt

Transaction Report:

January 13, 2021

RE: Life Science Alliance Manuscript #LSA-2020-00972-T

Ian Ahearn

Dear Dr. Ahearn,

Thank you for submitting your revised manuscript entitled "NRAS is Unique Among RAS Proteins in Requiring ICMT for Trafficking to the Plasma Membrane". Your manuscript has now been reviewed by a panel of 3 reviewers, whose reports are appended below.

As you will note from the reviewers' comments below, the reviewers have shown quite enthusiasm for these findings, and have suggested only minor changes to bring it to publication level. We encourage you to address all of the reviewers' points by text changes only, including some experimental requests made by reviewer 2 and 3. We would be happy to publish your paper in Life Science Alliance pending final revisions necessary to meet the reviewers' points and our formatting guidelines.

Along with the points listed below, please also attend to the following,

- please consult our manuscript preparation guidelines <https://www.life-science-alliance.org/manuscript-prep> and make sure your manuscript sections are in the correct order;
- please separate the Results and Discussion section into two - 1. Results 2. Discussion, as per our formatting requirements
- please add a Category and Summary blurb for your manuscript in our system
- please add an Author Contributions section to your main manuscript text and the system
- please upload your figures as single files
- please add a callout for Figure 5D to your main manuscript text
- please move your main figure legends in the manuscript text after the references
- please add a conflict of interest statement to your main manuscript text
- please upload your main manuscript text as an editable doc file
- please use the [10 author names, et al.] format in your references (i.e. limit the author names to the first 10)
- please revise the inset position in Figure 2D so that they match the zoomed-in parts
- please add scale bars to Fig. 1D, E, F, all panels in Fig 2, Fig 4A, C and Fig 5A, B
- please provide a high resolution image and the original unedited source data for the blots in Fig 1G, 3A and 3B

To avoid unnecessary delays in the acceptance and publication of your paper, please read the

following information carefully.

A. FINAL FILES:

B. MANUSCRIPT ORGANIZATION AND FORMATTING:

Sincerely,

Shachi Bhatt, Ph.D.
Executive Editor
Life Science Alliance
<https://www.lsjournal.org/>
Tweet @SciBhatt @LSAJournal

Reviewer #1 (Comments to the Authors (Required)):

The authors revisit the importance of the carboxymethylation that occurs to the C-terminal cysteine of all Ras isoforms. It is a reversible modification that occurs following proteolysis and farnesylation of Ras proteins. The precise sequence and contributions of this series of post-translational modifications of Ras have been elucidated in a series of high impact papers over the last 30 years. This has been important not just for understanding Ras biology, but also for the development of therapeutic strategies that interfere with the correct membrane binding and localisation of Ras proteins. This manuscript identifies a subtle but important difference in the dependence of one of the Ras isoforms on carboxymethylation.

Figure 1D provides unambiguous evidence for a selective requirement of carboxymethylation for correct localisation of NRAS but not the other 3 Ras isoforms. This is a really noteworthy result likely to be of broad interest to the Ras field.

The rest of the paper generates detailed understanding of where this requirement is specifically acting. It fairly concludes that carboxymethylation promotes palmitoylation (potentially via decreasing the rate of depalmitoylation). This has localisation and trafficking consequences by reducing membrane residency although it did not prevent Golgi recruitment necessary for palmitoylation the lack of carboxymethylation interfered with post-Golgi PDE6d mediated trafficking of NRAS to the plasma membrane.

This is a well presented manuscript that contains sufficient conditions and controls to support the conclusions. I don't have specific requests for further experiments that would substantially improve this study. Any extra experiments that I could suggest would be incremental or take the study in a new direction that would be unfair to these interesting core observations.

Reviewer #3 (Comments to the Authors (Required)):

The manuscript by Ian Ahearn and colleagues describes an unexpected and important finding regarding trafficking of NRAS protein to the plasma membrane in human cancer cell line systems. The authors are using relevant experiments and adequate techniques. Overall, the quality of the figures, imaging and pictures are high, and the conclusions backed up with data. However, some minor questions should be addressed before publication.

1. The authors need to motivate the choice of the melanoma cell line sk-mel-28, the main cancer cell line used throughout most of the manuscript. Sk-mel-28 is mutant for BRAF V600E but NRAS WT. Does the ICMT inhibition/deletion have the same effect on NRAS trafficking on melanoma cell lines carrying a NRAS mutation Even though it might seem obvious, it is relevant to determine

whether NRAS trafficking kinetics to the PM/Golgi remain the same for both WT and MUTANT NRAS.

2. The authors have included in the figure legends of Fig3 A SKMEL147 and Fig5 A SKMEL173, two melanoma cell lines that are mutant for NRAS (Q61K), however these cell lines seem to only be mentioned here and not in the methods section. The authors might have performed some complimentary experiments on these cell lines which would increase the value of the findings in the manuscript.

3. To validate the major findings in Figure 1, the authors use U-2-SO Osteosarcoma cell line, driven by mutations in NF1 and TP53, an important experiment suggesting that the NRAS trafficking findings seems independent of cell of origin, the authors should consider mentioning this in the results.

4. Fig1A: It seems that the total levels of NRAS are reduced in sgICMT cells. Can the authors comment on that? Perhaps the stability of endogenous NRAS is reduced because of mislocalized NRAS? This could be easily addressed with a CHX pulse experiment.

5. Fig1C: The authors should explain the use of the "SAAX" mutant in the result text that it cannot be prenylated.

6. Fig 2A. For clarity, the authors should explain in the results section that GalT is galactyltransferase expressed in the Golgi, and is used as a marker for Golgi localization.

7. The authors use the term silencing ICMT on multiple places, for CRISPR Cas9 knockout experiments. Silencing suggests RNA interference, the authors should adjust terminology.

8. The following text in the method should be adjusted for clarity: CRISPR/Cas9 Genomic disruption of ICMT was performed by infecting cells with lentivirus generated by transfecting HEK293 cells with pLentiCRISPR v2 with a sgRNA targeting exon 1 of ICMT (sgICMT, 5'-CACCGCACCGGGCTGGCGCTCTACG-3' and 5'-AAACCGTAGAGCGCCAGCCCGGTGC-3') and Cas9 using Lipofectamine 3000 (Invitrogen, Life Technologies). Control cells were generated by infecting SKMEL28 cells with lentivirus expressing an sgRNA targeting Tomato fluorescent protein (sgTom, 5'GCCACGAGTTCGAGATCGA and 5'-TCGATCTCGAACTCGTGGC) and Cas9. Cells were selected with 2 µg/ml puromycin 2 d after infection, and used immediately for experiments. Currently it reads like SKMEL28 cells that are sgTOM or sgICMT where generated very differently one involving HEK cells the other not.

9. The introduction reads well, however, the authors could consider shortening it a bit as it currently is relatively long containing six paragraphs.

10. A schematic figure about the different PTM steps and the inhibitors used in the study would help the reader to follow the result section more easily.

Reviewer #4 (Comments to the Authors (Required)):

Ahearn et al present new evidence for a cooperative role of palmitoylation and methylation in the PM localization of NRAS. The study is well controlled and thorough and the conclusions are justified. The work presents important new information on RAS trafficking. I have only a few suggestions that the authors may wish to consider to clarify and extend the inferences that they draw from their data.

1. Based on the multiple papers of the Bastiaens group the involvement of PDEdelta in anterograde trafficking of NRAS would imply also a role for the RE, in that Arl2 mediated release of RAS cargo in the vicinity of the RE is required for onward vesicular trafficking to the PM. In this context Misaki et al posited a role for the RE as a waystation for certain mono-palmitoylated RAS proteins (PMID20876282), with different outcomes depending on the location of the palmitoyl group. Did the

authors consider this model in interpreting their results? It would be useful to at least include some discussion along these lines and perhaps also examine experimentally the role of ICMT / PDEdelta and ARL2 in the trans RE trafficking of both mono-palmitoylated HRAS mutants, the 181S mutant (not shown) in addition to the 184S mutant. Not least because the Misaki study predated knowledge of PDEdelta. Such experiments would also resolve whether the different requirements for NRAS PM localization are simply due to mono-palmitoylation, or also the location of the palmitate and by inference the actual structure of the C-terminal anchor.

2. In this context observations from several groups over the years ago (papers from the current authors plus for example PMID20876282 / 16024806) have shown that the distribution of H181S and H184S mono-palmitoylated HRAS mutants are quite different, with the 181S showing more extensive Golgi and/or RE localization than 184S, which like NRAS has a mixed Golgi / PM distribution. In the light of experiments / discussion suggested in point 1, can the new study shed any light on this interesting cell biology?

3. A model / diagram summarizing the results and revised trafficking routes / membrane interactions that are now shown to be methylation / PDEdelta dependent, including the concepts discussed above would be a useful addition to the MS.

Reviewer #1 (Comments to the Authors (Required)):

The authors revisit the importance of the carboxymethylation that occurs to the C-terminal cysteine of all Ras isoforms. It is a reversible modification that occurs following proteolysis and farnesylation of Ras proteins. The precise sequence and contributions of this series of post-translational modifications of Ras have been elucidated in a series of high impact papers over the last 30 years. This has been important not just for understanding Ras biology, but also for the development of therapeutic strategies that interfere with the correct membrane binding and localisation of Ras proteins. This manuscript identifies a subtle but important difference in the dependence of one of the Ras isoforms on carboxymethylation.

Figure 1D provides unambiguous evidence for a selective requirement of carboxymethylation for correct localisation of NRAS but not the other 3 Ras isoforms. This is a really noteworthy result likely to be of broad interest to the Ras field.

The rest of the paper generates detailed understanding of where this requirement is specifically acting. It fairly concludes that carboxymethylation promotes palmitoylation (potentially via decreasing the rate of depalmitoylation). This has localisation and trafficking consequences by reducing membrane residency although it did not prevent Golgi recruitment necessary for palmitoylation the lack of carboxymethylation interfered with post-Golgi PDE6d mediated trafficking of NRAS to the plasma membrane.

This is a well presented manuscript that contains sufficient conditions and controls to support the conclusions. I don't have specific requests for further experiments that would substantially improve this study. Any extra experiments that I could suggest would be incremental or take the study in a new direction that would be unfair to these interesting core observations.

We thank you for your positive review of our work and are glad you found it interesting.

Reviewer #3 (Comments to the Authors (Required)):

The manuscript by Ian Ahearn and colleagues describes an unexpected and important finding regarding trafficking of NRAS protein to the plasma membrane in human cancer cell line systems. The authors are using relevant experiments and adequate techniques. Overall, the quality of the figures, imaging and pictures are high, and the conclusions backed up with data. However, some minor questions should be addressed before publication.

1. The authors need to motivate the choice of the melanoma cell line sk-mel-28, the main cancer cell line used throughout most of the manuscript. Sk-mel-28 is mutant for BRAF V600E but NRAS WT. Does the ICMT inhibition/deletion have the same effect on NRAS trafficking on melanoma cell lines carrying a NRAS mutation Even though it might seem obvious, it is relevant to determine whether NRAS trafficking kinetics to the PM/Golgi remain the same for both WT and MUTANT NRAS.

We decided to use SKMEL28 cells for the majority of the experiments in the manuscript due to the desirable morphology for imaging. GFP-NRAS can be clearly seen decorating the plasma membrane and golgi, thus allowing the effect of loss of ICMT to be unambiguously observed. During the course of the study we performed imaging experiments from figure 1D in multiple cell lines including SKMEL147 and SKMEL173, that as you mention below are mutant for NRAS, and found the effect of loss of ICMT on the distribution of GFP-NRAS to be the same. We decided not to include this data just to keep the manuscript as concise as possible. We have also examined the distribution of mutant GFP-NRAS12V in the presence and absence of ICMT and found it to be indistinguishable from wild type.

2. The authors have included in the figure legends of Fig3 A SKMEL147 and Fig5 A SKMEL173, two melanoma cell lines that are mutant for NRAS (Q61K), however these cell lines seem to only be mentioned here and not in the methods section. The authors might have performed some complimentary experiments on these cell lines which would increase the value of the findings in the manuscript.

We have amended the methods section to include information about these cell lines. As described above we performed the experiment in figure 1D in multiple cell lines and while we found the SKMEL28 cells to give the clearest images, the results were the same in all cell lines we looked at.

3. To validate the major findings in Figure 1, the authors use U-2-SO Osteosarcoma cell line, driven by mutations in NF1 and TP53, an important experiment suggesting that the NRAS trafficking findings seems independent of cell of origin, the authors should consider mentioning this in the results.

We have included a comment that highlights this observation in the results section.

4. Fig1A: It seems that the total levels of NRAS are reduced in sgICMT cells. Can the authors comment on that? Perhaps the stability of endogenous NRAS is reduced because of mislocalized NRAS? This could be easily addressed with a CHX pulse experiment.

Yes, we also noted this interesting observation and did attempt to elucidate the mechanism. We performed CHX pulse chase experiments as suggested and found no significant difference in the stability of NRAS protein in the absence of methylation by ICMT. However, we did observe a decrease in the abundance of NRAS mRNA in the ICMT deficient cells. While fascinating we believe a full work up on this finding to be outside the scope of the study in this manuscript.

5. Fig1C: The authors should explain the use of the "SAAX" mutant in the result text that it cannot be prenylated.

We have included a sentence that clarifies this.

6. Fig 2A. For clarity, the authors should explain in the results section that GalT is galactyltransferase expressed in the Golgi, and is used as a marker for Golgi localization.

We have done this.

7. The authors use the term silencing ICMT on multiple places, for CRISPR Cas9 knockout experiments. Silencing suggests RNA interference, the authors should adjust terminology.

We have changed the terminology to “disruption” instead of “silencing” to avoid confusion.

8. The following text in the method should be adjusted for clarity: CRISPR/Cas9 Genomic disruption of ICMT was performed by infecting cells with lentivirus generated by transfecting HEK293 cells with pLentiCRISPR v2 with a sgRNA targeting exon 1 of ICMT (sgICMT, 5'-CACCGCACCGGGCTGGCGCTCTACG-3' and 5'-AAACCGTAGAGCGCCAGCCCGGTGC-3') and Cas9 using Lipofectamine 3000 (Invitrogen, Life Technologies). Control cells were generated by infecting SKMEL28 cells with lentivirus expressing an sgRNA targeting Tomato fluorescent protein (sgTom, 5'GCCACGAGTTCGAGATCGA and 5'-TCGATCTCGAACTCGTGGC) and Cas9. Cells were selected with 2 µg/ml puromycin 2 d after infection, and used immediately for experiments.

Currently it reads like SKMEL28 cells that are sgTOM or sgICMT where generated very differently one involving HEK cells the other not.

This has been amended to show that HEK293 cells were used to generate both sgTOM or sgICMT CRISPR lentivirus.

9. The introduction reads well, however, the authors could consider shortening it a bit as it currently is relatively long containing six paragraphs.

We have noted this suggestion but found any attempts to shorten the introduction resulted in a loss of clarity so we elected to keep it at the current length.

10. A schematic figure about the different PTM steps and the inhibitors used in the study would help the reader to follow the result section more easily.

We feel that our results highlight the complexity of both interdependent and independent effects of multiple PTM steps which make it difficult to depict in a schematic that is simple yet accurate and so we prefer not to do so here.

Reviewer #4 (Comments to the Authors (Required)):

Ahearn et al present new evidence for a cooperative role of palmitoylation and methylation in the PM localization of NRAS. The study is well controlled and thorough

and the conclusions are justified. The work presents important new information on RAS trafficking. I have only a few suggestions that the authors may wish to consider to clarify and extend the inferences that they draw from their data.

1. Based on the multiple papers of the Bastiaens group the involvement of PDEdelta in anterograde trafficking of NRAS would imply also a role for the RE, in that Arl2 mediated release of RAS cargo in the vicinity of the RE is required for onward vesicular trafficking to the PM. In this context Misaki et al posited a role for the RE as a waystation for certain mono-palmitoylated RAS proteins (PMID20876282), with different outcomes depending on the location of the palmitoyl group. Did the authors consider this model in interpreting their results? It would be useful to at least include some discussion along these lines and perhaps also examine experimentally the role of ICMT / PDEdelta and ARL2 in the trans RE trafficking of both mono-palmitoylated HRAS mutants, the 181S mutant (not shown) in addition to the 184S mutant. Not least because the Misaki study predated knowledge of PDEdelta. Such experiments would also resolve whether the different requirements for NRAS PM localization are simply due to mono-palmitoylation, or also the location of the palmitate and by inference the actual structure of the C-terminal anchor.

We are familiar with the interesting results from Misaki et al., and this work did indeed guide aspects of our own investigative efforts. We now include a citation for this excellent work in our discussion. We have performed the experiment to test the impact of Arl2 silencing on the trafficking of GFP-NRAS in the presence or absence of ICMT inhibition. Our findings did not convincingly demonstrate any discreet phenotype or definitive modulation of the effect of ICMT loss. We did not test this with individual palmitoylation mutants of HRAS, though we agree this may reveal insight for how palmitoylation contributes to PDE6 δ trafficking with respect to the recycling endosomal compartment. Despite this, including evidence in this context would not, we feel, significantly refine the focus of our work on the role of carboxylmethylation for NRAS with our much additional effort at this time.

2. In this context observations from several groups over the years ago (papers from the current authors plus for example PMID20876282 / 16024806) have shown that the distribution of H181S and H184S mono-palmitoylated HRAS mutants are quite different, with the 181S showing more extensive Golgi and/or RE localization than 184S, which like NRAS has a mixed Golgi / PM distribution. In the light of experiments / discussion suggested in point 1, can the new study shed any light on this interesting cell biology?

We did not individually interrogate the impact of ICMT loss on the distribution of HRAS181S and, therefore, any insight we can draw is limited with regard to the contribution of the two palmitoylations sites for Golgi/recycling endosomal trafficking.

3. A model / diagram summarizing the results and revised trafficking routes / membrane interactions that are now shown to be methylation / PDEdelta dependent, including the concepts discussed above would be a useful addition to the MS.

Though we believe we can confidently attribute a unique dependence of ICMT for NRAS PM trafficking due to changes in palmitoylation and PDE6 δ binding, we feel it that more refinement is needed before an updated integrated model of NRAS trafficking is put forth in a schematic depiction.

January 27, 2021

RE: Life Science Alliance Manuscript #LSA-2020-00972-TR

Ian M Ahearn
NYU Langone Medical Center and NYU Grossman School of Medicine
The Ronald O. Perelman Department of Dermatology and The Perlmutter Cancer Center
522 First Ave
Smilow Research Building, Fl 12
New York, NY 10016

Dear Dr. Ahearn,

Thank you for submitting your Research Article entitled "NRAS is Unique Among RAS Proteins in Requiring ICMT for Trafficking to the Plasma Membrane". It is a pleasure to let you know that your manuscript is now accepted for publication in Life Science Alliance. Congratulations on this interesting work.

DISTRIBUTION OF MATERIALS:

Again, congratulations on a very nice paper. I hope you found the review process to be constructive and are pleased with how the manuscript was handled editorially. We look forward to future exciting submissions from your lab.

Sincerely,

Shachi Bhatt, Ph.D.

Executive Editor

Life Science Alliance

<https://www.lsjournal.org/>
